

# Deniable ring signature scheme based on the ISRSAC digital signature algorithm

Yanshuo Zhang, Yuqi Yuan, Ning Liu, Ying Chen and Youheng Dong

Department of Cryptology Science and Technology, Beijing Electronic Science & Technology Institute, Beijing, China

## ABSTRACT

This article introduces a deniable ring signature scheme within the framework of a ring signature protocol, which allows a member of the ring to assert or deny their role as the signature's originator without the involvement of a trusted third party. Deniable ring signatures find applications in various scenarios where balancing privacy protection with supervised oversight is crucial. In this work, we present a novel deniable ring signature scheme based on the ISRSAC digital signature, tailored for environments with regulatory requirements, empowering regulators to hold malicious signers accountable. We formally validate the scheme's correctness, unforgeability, anonymity, traceability, and non-frameability through rigorous proofs. Additionally, we conduct a comparative analysis of the communication and computational overheads associated with our proposed scheme, demonstrating its superiority in practical scenarios such as electronic auctions and anonymous reporting mechanisms.

## INTRODUCTION

A cryptographic notion called a ring signature is employed in digital signatures to safeguard signers' identity and privacy. A typical digital signature uses the signer's private key to certify the data, which the public key may confirm. Ring signatures, on the other hand, make it possible to sign a group of people without disclosing who each participant is. Every participant in a ring signature possesses a pair of keys, a public key and a private key. Members select a ''ring'' or group of members to sign with, then use their unique private keys to sign the data. The buyer can verify that the signature is genuine, but it is impossible to ascertain whose member signed it. One salient characteristic of ring signatures is the arbitrary selection of member sets; furthermore, it is impossible to identify the real signer of a member collection including many members at the same time. This makes ring signatures ideal for situations requiring privacy protection and anonymity, such anonymous authentication, anonymous voting, and so on. *Rivest, Shamir & Tauman (2001)* proposed ring signatures to achieve digital signatures that are anonymous to identity. *Bresson, Stern & Szydlo (2002)* proposed the first t-out-of-N gate ring signature. In order to make it possible for anyone to ascertain if two ring signatures were created by the same signer, *Tsang et al. (2004)* presented a separable linkability threshold ring signature system. In order to meet

Corresponding author
Yuqi Yuan, 18812518820@163.com

the safety criteria in real-world applications (*Xuemin et al., 2024*; *Xie et al., 2024*; *Li et al., 2024*), researchers have now carried out more research and created ring signature schemes with various features.

In order to validate who the true signer of the ring signature is, *Komano et al. (2006)* introduced a confirmation/denial algorithm to the standard ring signature and presented the idea of a negated ring signature. Deniable ring signatures are traceable and appropriate for application environments requiring supervision because they can trace the true signer without involving a third party, in contrast to the unconditional anonymity of general ring signatures, which prevents regulators from holding malicious signers accountable. *Wu et al. (2006)* used accumulators and knowledge signatures to generate self-organizing group signatures that were demonstrably safe and repudiable. *Hu & Liu (2011)* combined the concepts of "constant-size" and "deniable" to form an identity-based deniable ring signature with a constant-size signature. They proposed a new scheme with constant-size signature length based on an improved accumulator from bilinear pairings, which resolves the issue of anonymity abuse. Using the security paradigm put forth by *Komano et al. (2006)*, *Zeng, Jiang & Qin (2012)* created a deniable ring signature based on bilinear pairings. This signature can demonstrate its security under the random oracle machine model, where the confirmation/denial mechanism operates in a non-interactive manner. According to the standard paradigm, *Zeng et al. (2013)* suggests a non-interactive deniable ring signature system that can demonstrate security.

Better security performance under public-key cryptography is achieved *via* the ISRSAC algorithm, which is an enhancement of the classic RSA method. In addition, as it is the fundamental algorithm for creating digital signature schemes, digital signatures built using ISRSAC have more security than those built using RSA. *Thangavel et al. (2015)* suggested the ESRKGS method, which makes breaking apart big numbers more challenging. *Yang et al. (2021)* created a digital signature system based on ISRSAC, and they created a broadcast multi-signature scheme, an ordered multi-signature scheme, and a proxy signature scheme using this foundation. A sequential proxy multi-signature scheme based on ISRSAC and broadcast proxy multi-signature techniques was devised by *Liu, Zhang & Ning (2022)*, building on *Yang et al. (2021)*. They built a ring signature system using the ISRSAC algorithm and demonstrated several security features. An adaptor signature system based on ISRSAC was presented by *Zhang et al. (2023)* in 2023 and has a lot of potential applications in blockchain and other use cases. Since then, *Liu, Yuan & Zhang (2023)* have suggested a study review and carried out extensive research on the ISRSAC algorithm and the digital signature scheme created based on the method. These days, an increasing number of academics have built digital signature schemes based on ISRSAC that are appropriate for many circumstances, coupled digital signature technology with the security benefits of the algorithm, and researched and debated its applicability in numerous industries.

One disadvantage of ring signatures is that their ability to provide anonymity can be exploited by a malicious signer, which can then raise suspicions about the other members of the ring. Deniable ring signatures were created for this purpose. Initially, it can protect the person signing, enable other members of the group to deny their participation, and

offer proof of the actual signer's identity. These types of signatures have numerous practical applications.

Even though the deniable ring signature system is really helpful, there are still several issues with it as it is. It is clearly unfair to honest ring members for attackers to be able to forge denials on behalf of some of them due to specific restrictions in various schemes' security models and failure to take non-frameability into consideration. Second, for the majority of schemes, the size of the rejections and signatures grows with the square of the ring membership. This implies that the size of signatures and rejections rises considerably with the number of ring members. Consequently, another pressing issue is how to create a sublinear deniable ring signature method. To further address the aforementioned issues, this article suggests a deniable ring signature scheme based on ISRSAC. This scheme effectively lowers traffic volume, enhances security at the signature algorithm level, improves security model-based security, and increases operation efficiency through better implementation.

## BACKGROUND

### Difficult problem of factoring large integers

The decomposition of a large integer as the product of its prime factors, or the representation of a large integer as the product of two or more prime numbers, is known as the difficulty factoring large integers problem. The description of the problem can be simply expressed as: given a large integer $n$, find two prime numbers $p, q$ such that $n = p \cdot q$. In this problem, the size of $n$ is usually very large, far beyond what a computer can solve in a reasonable amount of time. The problem cannot be solved efficiently using a polynomial algorithm. This means that extracting its prime factor from a large integer requires a considerable number of computational resources and time. Therefore, the problem is considered "difficult".

The hard challenge of massive integer factorization serves as the foundation for the RSA encryption technique. The intricacy of factoring a big integer into the product of two prime integers forms the foundation of the security of the RSA encryption technique. A major concern to contemporary computer security is the possibility of cracking communication encrypted using RSA if someone can successfully solve the huge integer factorization problem.

Although the difficult problem of large integer factorization is theoretically considered a difficult problem, there are still algorithms that can be used to solve large integer factorization in some specific cases. For example, prime factorization algorithms, such as Pollard's rho algorithm, the secondary sieving method, and the number field sieving method, can efficiently decompose large integers in certain situations. However, these algorithms typically only operate with a limited range of integers. For larger integers, they become time-consuming and impractical.

### ISRSAC digital signature algorithm

This section briefly introduces the digital signature algorithm based on ISRSAC, which includes four PPT algorithms: System initialization (Setup), key generation algorithm (KeyGen), signature generation algorithm (Sign), and signature verification algorithm (Verify).

**Table 1  Symbol description.**

| Symbol | Description |
| --- | --- |
| List | A list of public keys published by PKI<br>A list of malicious signers that have been compromised<br>or registered by an attacker. |
| MList | We call signers honest signers who are not in MList<br>a list of message signature pairs generated by the challenge<br>oracle query $CH_b(.,.,.)$ |
| Gset | Note that an attacker cannot query the<br>message signature pairs contained in<br>GSet through the oracle machine $C/D(.,.,.)$ to<br>acknowledge/deny |
| ⊥ | the algorithm abort |
| \ | the difference between two sets |
| Ø | empty set |
| $\lambda$ | Security parameters |
| PPT | Probabilistic polynomial time |
| $\Gamma$ | A group of elliptic curve points of order $q$ |
| $G$ | A generator of $\Gamma$ |
| $H_1:\{0, 1\}^n \to Z_q^*$ | Hash algorithm |
| $H_2:\{0, 1\}^n \to G$ | Hash algorithm |

1. System initialization (Setup): given the safety parameters $\lambda$, randomly select two large prime numbers $p$ and $q$, where $p \neq q$, $p > 3$, $q > 3$, calculate $n = p \cdot q \cdot (p-1) \cdot (q-1)$, $m = p \cdot q$. Randomly select an integer $r$, where this condition of $r$ is met: $p > 2^r < q$, and then calculate $\alpha(n) = \frac{(p-1)(q-1)(p-2^r)(q-2^r)}{2^r}$.

2. Key generation algorithm (KeyGen): Select the public key $e$, where $1 < e < \alpha(n)$, $gcd(e, \alpha(n)) = 1$. Calculate the private key $d$, where $d \cdot e \equiv 1 (\bmod \alpha(n))$. In summary, determine the public key $(e, m)$ and private key $(d, n)$.

3. Signature generation algorithm (Sign): Suppose the plaintext is $M$, the hash function is $H$, and the hash value is $H(M)$. The private key is $(d, n)$, calculated $S \equiv H(M)^d (\bmod m)$ as the signature of $H(M)$.

4. Signature verification algorithm (Verify): Verify the signature $S$ on the message $M$, compute $H(M)$ using the same hash function, and use public key $(e, m)$ to verify if $H(M) = S^e (\bmod m)$ is correct. If the equation is correct, then the signature is accepted.

## Security model

In this subsection, we describe the Oracle machines used in the security model. The following symbol descriptions are given in Table 1.

1. Join Oracle $(Add(ID_i) \to pk_i)$: Call Join Oracle Machine in order to add the identification of the truthful signer $ID_i$ to the List. The oracle machine returns $\epsilon$ if there is already a signer with identification $ID_i$. If not, a key-generating process is executed by the oracle machine, which appends the signer (together with its public key) to the List. At last, $pk_i$ is returned by the oracle machine.

2. Register oracle machine $(Reg(ID_i, pk_i))$: An attacker can register a new signer with the public key $pk_i$ in the List by using a signer to register an oracle. Signers were also added to MList by the oracle machine.

3. Corrupt Oracle $(Crpt(ID_i, pk_i) \rightarrow sk_i)$: Corrupting the signatories of the identity $ID_i$ by use of a corrupt oracle. The key $sk_i$ of signer $P_i$ can be extracted by an attacker from the oracle.

4. Signature oracle machine $(DRSig(ID_k; M, ID_1, \ldots, ID_{k-1}, ID_{k+1}, \ldots, ID_{L'}, List) \rightarrow \sigma)$: The signature $\sigma$ of the honest signer and the entity is output given the identity of the honest signer $P_{i_k}$, message $M$, and entity $P_{i_1}, \ldots, P_{i_{k-1}}, P_{i_{k+1}}, \ldots, P_{L'}$ of the signature oracle.

5. Challenge Oracle $(Ch_b(ID_0, ID_1, M) \rightarrow \sigma)$: Anonymity is defined as using a challenge machine. Given the identity $(i_0, i_1)$ and message $M$, the oracle machine provides the target signature $S(sk_{i_0}, M, pk_{i_0}, pk_{i_1})$ from $P_{i_0}$ and $P_{i_1}$ for $M$ for a challenge bit $b \in 0, 1$. Keep in mind that attackers are not limited to weakening signers $P_{i_0}$ and $P_{i_1}$. Furthermore, the target's signature would be protected against an attacker's ability to execute the $P_{i_0}$ and $P_{i_1}$ confirmation/denial protocols. The attacker challenges the oracle machine to add the target signature to the GSet in the event that it is confirmed or denied.

6. Confirm/Deny Oracle $(C/D(ID_i, M, \sigma))$: Given an identity $ID_i$ and a message signature pair $(M, \sigma)$, a confirm/deny oracle verifies if $P_i$ is an honest signer by executing the confirm/deny protocol and communicating with the attacker. An oracle does not execute the confirmation/denial procedure if an attacker inserts a target signature that questions the results.

On the basis of the mentioned above, we improve the security model mentioned above and provide the necessary characteristics of a deniable ring signature, which are traceability, non-frameability, accuracy, and anonymity.

## OUR PROPOSED SIGNATURE SCHEME

### Deniable ring signature

**Definition 1. Deniable Ring Signature Scheme**

The member (victim) whose public key is used to create the ring signature in a deniable ring signature scheme may assert that false allegations were made against them. Without a group administrator, the scheme is a group signature scheme. To represent the quantity of signers in the system, we use $L$. According to the definition that follows, the signer $P_{i_k}$ dynamically chooses $L' - 1$ entities $P_{i_1}, \ldots, P_{i_{k-1}}, P_{i_{k+1}}, \ldots, P_{i_{L'}}$ from the signer list that PKI publishes, and then performs the signature algorithm by entering their own private key and the entity's public key collection.

The general construction algorithm for a deniable ring signature is as follows:

1. System initialization (Setup $(\lambda) \rightarrow$ params): Given the security parameters $\lambda$, output the system parameters params.

2. Key generation (KeyGen $(params \rightarrow (pk_i, sk_i))$): Output the public–private key pair $(pk_i, sk_i)$ of user $A_i$ given the system parameters params.

3. Signing (Sign $(m, sk_i, L) \to \sigma$): Given the message m, the user $A_i's$ $(1 \leq i \leq n)$ private key $sk_i$, and public key list $L = \{pk_1, \ldots, pk_i, \ldots, pk_n\}$, output the signature value. Among them, the public key list $L$ is composed of randomly selecting the $n-1$ users' public key for the user, plus its own public key $pk_i$, that is $L = \{pk_1, \ldots, pk_i, \ldots, pk_n\}$.

4. Verification (Verify $(m, \sigma, L) \to accept/reject$): given the message $m$, signature value $\sigma$, public key list $L$, if $\sigma$ is a legal signature, output "accept"; otherwise, output "reject".

5. Confirmation (Confirm: $P \Leftrightarrow V$): The interaction algorithm between the prover $P$ (*i.e.,* signer $A_i$) and the verifier $V$, the verifier $V$ has a message $m$, signature value $\sigma$, public key list $L$, and the prover $P$ proves that the ring signature was generated by it through the confirmation algorithm.

6. Disavowal (Disavow: $P \Leftrightarrow V$): An algorithm for the interaction between the prover $P$ (non-signer $A_j, j \neq i$) and the verifier $V$, the verifier $V$ has a message $m$, signature value $\sigma$, public key list $L$, and the prover $P$ denies that the ring signature was generated by the disavowal algorithm.

## Security properties

The security model proposed in *Komano et al. (2006)* gives the required properties of a deniable ring signature: correctness, unforgeability, anonymity, traceability, and non-frameability.

**Definition 2. Correctness:** We argue that the disavowal ring signature is correct, and the confirmation/disavowal algorithm may identify the signer of the signature if the signature generated by the right signature algorithm is approved by the verification method. With respect to adversary $A$, safety parameters $k$, and experiment $Exp_{DRS,A}^{corr}(k)$, we formally characterize the correctness of the deniable ring signature as follows:

1. Initialize the list: $List \leftarrow \emptyset, MList \leftarrow \emptyset, GSet \leftarrow \emptyset$.

2. Adversary $A$ inquires the oracle machine:
   $(ID_k; M; ID_1, \ldots, ID_{L'}) \leftarrow A(1^k, Add(\cdot), Reg(\cdot, \cdot), Crpt(\cdot), C/D(\cdot, \cdot, \cdot), DRSign(\cdot; \cdot, \cdot))$.

3. If $\{ID_1, ID_2, \ldots, ID_i, \ldots, ID_n\} \notin List \setminus MList$, then return 1.

4. Simulator calculations $\sigma \leftarrow S(sk_{i_k}, M, pk_{i_1}, \ldots, pk'_{i_L})$, if verified $V(M, \sigma, pk_{i_1}, \ldots, pk_{i'_L}) = 0$, then return 1.

5. If the returned disavowal algorithm from $C/D(ID_k, M, \sigma)$ executed successfully, then return 1.

6. If the returned confirmation algorithm from $C/D(ID_j \neq ID_i, M, \sigma)$ executed successfully, then return 1.

7. In other cases, return 0.

The definition of adversary $A$'s advantage is as follows: $Adv_{\Sigma,A}^{corr} = Pr[Exp_{\Sigma,A}^{corr} = 1]$. The deniable ring signature strategy is correct if the advantage $Adv_{\Sigma,A}^{corr}$ is small compared to any PPT opponent $A$.

**Definition 3. Unforgeability:** Through the experiments between simulator $S$ and PPT adversary $A$ to formally define unforgeability. First, the simulator $S$ generates the system parameter and sends it to the adversary, then initializes each list, $A$ adaptive query oracle, output message, public key list, and forged signatures, and then passes through the verification function to verify.

Formal definition: For a deniable ring signature, an arbitrary adversary $A$, define the experiment $Exp_{\Sigma,A}^{euf}$ as follows:

1. Initialize the list: $List \leftarrow \emptyset$, $MList \leftarrow \emptyset$, $GSet \leftarrow \emptyset$.
2. Adversary A asks the oracle for messages, public keys, and forged signatures: $(M, \sigma^*, pk_{i_1}, \ldots, pk_{i'_L}) \leftarrow A(1^k, Add(\cdot), Reg(\cdot, \cdot), Crpt(\cdot), C/D(\cdot, \cdot, \cdot), DRSign(\cdot; \cdot, \cdot))$ and $\sigma^*$ is not obtained by querying the signature oracle machine $DRSign(\cdot; \cdot, \cdot)$.
3. For all $i \in [1, n]$, the identity $ID_i \notin List \setminus MList$ corresponding to $pk_i$.
4. If $V(M, \sigma, pk_{i_1}, \ldots, pk_{i'_L}) = 1$, then return 1; otherwise, return 0.

The opponent $A$'s advantage is defined as follows: $Adv_{\Sigma,A}^{euf} = Pr[Exp_{\Sigma,A}^{euf} = 1]$. For every PPT opponent $A$, the deniable ring signature method is unforgeable if the advantage $Adv_{\Sigma,A}^{euf}$ is insignificant.

**Definition 4. Anonymity:** Through the experiments between simulator $S$ and PPT adversary $A$ to formally define unforgeability. The simulator $S$ generates the system parameter and sends it to the adversary, then initializes each list, $A$ adaptive query oracle, outputs a bit $d$.

Formal definition: We describe the experiment $Exp_{DRS,A}^{anon-b}(k)$ for a deniable ring signature, an arbitrary adversary $A$, a bit $b \in 0, 1$, and a security parameter $k$ as follows:

1. Initialize the list: $List \leftarrow \emptyset$, $MList \leftarrow \emptyset$, $GSet \leftarrow \emptyset$.
2. Adversary $A$ inquires the oracle machine, outputs a bit $d \in 0, 1$, return $d$: $d \leftarrow A(1^k, Ch_b(\cdot, \cdot, \cdot), Add(\cdot), Reg(\cdot, \cdot), Crpt(\cdot), C/D(\cdot, \cdot, \cdot), DRSign(\cdot; \cdot, \cdot))$.

Defining the advantages of an adversary $A$:
$Adv_{\Sigma,A}^{anon} = \left| Pr[Exp_{\Sigma,A}^{anon-1} = 1] - Pr[Exp_{\Sigma,A}^{anon-0} = 1] \right| = \left| 2Pr[Exp_{\Sigma,A}^{anon-b} = b] - 1 \right|$.

When the deniable ring signature technique is anonymous, it means that the advantage $Adv_{\Sigma,A}^{anon}$ is negligible in comparison to any PPT opponent $A$.

**Definition 5. Traceability:** Traceability means that a real signer of a deniable ring signature can be traced by a confirmation/denial algorithm, through the experiments between simulator $S$ and PPT adversary $A$ to formally define unforgeability. The simulator $S$ generates the system parameter and sends it to the adversary, then initializes each list, $A$ adaptive query oracle, output message, public key list and signatures, and then passes through the verification function to verify.

Formal definition: For a deniable ring signature, arbitrary adversary $A$, and security parameters $k$, define the experiment $Exp_{DRS,A}^{trace}(k)$ shown as follows:

1. Initialize the list: $List \leftarrow \emptyset$, $MList \leftarrow \emptyset$, $GSet \leftarrow \emptyset$.
2. Adversary $A$ inquires the oracle machine, outputs messages, public key lists, and signatures: $(M, \sigma, pk_{i_1}), \ldots, pk_{i'_L} \leftarrow A(1^k, Add(\cdot), Reg(\cdot, \cdot), Crpt(\cdot), C/D(\cdot, \cdot, \cdot), DRSign(\cdot; \cdot, \cdot))$.
3. If $V(M, \sigma, pk_{i_1}, \ldots, pk_{i'_L}) = 0$, then return 0.
4. If users $P_{i_1}, \ldots, P_{i'_L}$ can deny $(M, \sigma)$, then return 1; otherwise, return 0.

Define the advantage of the adversary $A$: $Adv_{\Sigma,A}^{trace} = Pr[Exp_{\Sigma,A}^{trace} = 1]$. If the advantage $Adv_{\Sigma,A}^{trace}$ is negligible against any PPT adversary $A$, then the deniable ring signature scheme is traceable.

**Definition 6. Non-frameability:** Non-frameability means that adversary $A$ enables non-signers to deny signatures and ensures that adversary $A$ cannot frame real signers, through

the experiments between simulator $S$ and PPT adversary $A$ to formally define unforgeability. The simulator $S$ generates the system parameter and sends it to the adversary, then initialize each list, $A$ adaptive query oracle, output message, public key list and signatures, and then through the verification function to verify.

Formal definition: For a deniable ring signature, arbitrary adversary $A$, and security parameters $k$, define the experiment $Exp_{DRS,A}^{nf}(k)$ shown as follows:

1. Initialize the list: $List \leftarrow \emptyset$, $MList \leftarrow \emptyset$, $GSet \leftarrow \emptyset$.
2. Adversary $A$ inquires the oracle machine, outputs messages, public key lists, and signatures: $(M, \sigma, pk_{i_1}, \ldots, pk_{i'_L}) \leftarrow A(1^k, Add(\cdot), Reg(\cdot, \cdot), Crpt(\cdot), C/D(\cdot, \cdot, \cdot), DRSign(\cdot; \cdot, \cdot))$.
3. If $V(M, \sigma, pk_{i_1}, \ldots, pk_{i_{L'}}) = 0$, then return 0.
4. If the following conditions are met, return 1; otherwise, return 0.
   (a) The existence of user $P_{i_t}$ cannot be denied $(M, \sigma)$, $t \in [1, L]'$, *i.e.*, user $P_{i_t}$ is not the signer.
   (b) Adversary $A$ has not been queried $Crpt(ID_t)$, or $DRSign(ID_t; M, ID_1, \ldots, ID_{t-1}, ID_{t+1}, \ldots, ID_L)$.

Define the advantage of the adversary $A$: $Adv_{\Sigma,A}^{nf} = Pr[Exp_{\Sigma,A}^{nf} = 1]$. If the advantage $Adv_{\Sigma,A}^{nf}$ is negligible against any PPT adversary $A$, then the deniable ring signature scheme is not defamatory.

## Scenario description

This section will introduce the design of a deniable ring signature scheme based on ISRSAC, which consists of six PPT algorithms.

1. System initialization (Setup $(\lambda) \to params$): Given the security parameters $\lambda$, output the system parameter params.
2. Key generation (KeyGen $(params) \to (pk; sk)$): Given the system parameter params, the user in the ring performs the following steps:
   (a) Select two large integers $p$ and $q$, where $p > 3, q > 3$, generate $n = p \cdot q \cdot (p-1) \cdot (q-1)$, $m = p \cdot q$.
   (b) Randomly select an integer $r$ that meets the conditions $p > 2^r < q$ and generate $\alpha(n) = ((p-1)(q-1)(p-2^r)(q-2^r))/2^r$.
   (c) Select the public key index $e$ to satisfy $1 < e < \alpha(n)$ and $gcd(e, \alpha(n)) = 1$.
   (d) Calculate the private key index $d$, meet the condition: $e \cdot d \equiv 1 (\text{mod } \alpha(n))$.
   (e) Output the public–private key pair $pk = (e, n)$, $sk = (d, m)$.
3. Signature generation (Sign $(m; sk_i; L) \to sig$): Given the message $M$, the private key $sk_i$ of the signer $A_i$, the signer forms a list $L = pk_1, \ldots, pk_n$ of public keys through $n-1$ random users' public keys and their own private keys, hash function $H(M) : [0, 1]^* \to Z_m$, perform the following steps:
   (a) User $A_i$ calculates $S_i = H(M)^{d_i} \cdot \prod_{j=1, j \neq i}^{n} H(M)^{-e_j^2} (\text{mod } n)$.
   (b) The other ring members calculate $S_j = H(M)^{e_i e_j} (\text{mod } n)$, where $j = 1, 2, \ldots, i, \ldots, n$.
   (c) Output $(S_1, S_2, \ldots, S_n)$ about the ring signature of $M$.
4. Verification (Verify $(m; sig; L) \to (accept/reject)$): The verifier receives the message $M'$ and the ring signature value $(S'_1, \ldots, S'_n)$, and execute the following steps:
   (a) Calculate the value of $H(M')$.

(b) Calculate the value of $\prod_{j=1}^{n} S_j'^{e_j}$.

(c) Determine whether there is $\prod_{j=1}^{n} S_j'^{e_j} \equiv H(M') \pmod{n}$. If satisfied, output "accept"; otherwise, output "reject".

5. Confirmation (Confirm: $(P \Leftrightarrow V)$: An interaction algorithm between the prover $P$ (*i.e.,* signer) and the verifier $V$ that the prover $P$ can use to prove that it generated the corresponding ring signature. The verifier $V$ has a message $M$, signature value $(S_1', S_2', \ldots, S_n')$, and public key list $L$, and the prover $P$ proves through a confirmation algorithm that the ring signature was generated by it. The confirmation steps are as follows:

(a) $P \rightarrow V$: The prover selects $t \in Z_q^*$ randomly, calculate $R = H(M)^t \pmod{n}$, and sent $R$ to the verifier.

(b) $V \leftarrow P$: The validator picks $u \in Z_q^*$ at random and sends $u$ to the prover.

(c) $P \rightarrow V$: The prover calculates $w = t - u \cdot d_i \pmod{n}$ and sends $w$ to the verifier, where $d_i = \log_{H(M)} \frac{S_i}{\prod_{j=1,j \neq i}^{n} H(M)^{-e_j^2}} \pmod{n}$. The verifier verifies whether $R = H(M)^w \left( \frac{\prod_{j=1,j \neq i}^{n} H(M)^{-e_j^2}}{A} \right)^u \pmod{n}$ is valid, and if so, determines that the signature is signed by the prover.

6. Disavowal (Disavow: $(P \Leftrightarrow V)$: The disavowal algorithm is an interactive algorithm between the prover $P(non - signer A_j, j \neq i)$ and the verifier $V$, which the prover $P$ can use to deny that a ring signature has ever been generated. The algorithm requires the participation of three parties, three transmissions, and is implemented using the public key system. The unique identity of the user $A_j$ is $I_{A_j}$ transformed by the hash function to obtain the corresponding hash value $J_{A_j} = H(I_{A_j})$, and the trusted arbitrator $T$ assigns the key function $S_{A_j} = (H_{A_j})^{-d_j} \pmod{n}$ to $A_j$. The negative steps as follows:

(a) $P \rightarrow V$: The prover $P$ chooses a random number $r$, $1 \leq r \leq n - 1$, calculates $x = r^e \pmod{n}$, where $r$ is the secret random number chosen by $P$, $P$ sends $(I_{A_j}, x)$ to $V$.

(b) $V \leftarrow P$: The verifier $V$ selects the random number $u$, $1 \leq u \leq e$, and sends $u$ to $P$.

(c) $P \rightarrow V$: The prover $P$ calculate $y = r \cdot S_{A_j}^u \pmod{n}$, send to the verifier $V$, after $V$ receives $y$, $V$ calculates $I_{A_j}$ from $J_{A_j} = H(I_{A_j})$ and $J_A^u \cdot Y^e \pmod{n}$, if the result is not equal to $x$, then the signature is considered not signed by the prover.

## PROOF OF SECURITY

This section provides proof of the properties of the deniable ring signature based on ISRSAC.

### Theorem 1. The deniable ring signature based on ISRSAC satisfies correctness

**Proof**: Assuming that the disavowal ring signature value of the generated message $M$ is $\sigma = (S_1, \ldots, S_n)$, the correctness of the verification, confirmation and disavowal algorithms of the signature scheme were discussed below.

**The correctness of the verification algorithm:** Suppose the received message is $M'$, whose ring signature $(S'_1, \ldots, S'_n)$ is composed of the $S'_i$ through the $H_1(\cdot)$ operation in the signature generation algorithm, according to the equation $\prod_{i=1}^{n} S'^{e_i}_i = S'^{e_\pi}_\pi \cdot \prod_{j=1,j\neq\pi}^{n} S'^{e_i}_i$, where $S'_\pi = H_1(M')^{d_\pi} \cdot \prod_{i=1,i\neq\pi}^{n} H_1(M')^{-e_i^2} (\bmod\ n)$ and $\prod_{i=1,i\neq\pi}^{n} S'^{e_i}_i = \prod_{i=1,i\neq\pi}^{n} H_1(M')^{e_\pi e_i^2} (\bmod\ n)$. Then we have: $\prod_{i=1}^{n} S'^{e_\pi}_i = S'^{e_\pi}_\pi \cdot \prod_{j=1,j\neq\pi}^{n} S'^{e_j}_j = H_1(M')^{d_\pi e_\pi} \cdot \prod_{i=1,i\neq\pi}^{n} H_1(M')^{-e_\pi e_i^2} (\bmod\ n) = H_1(M')(\bmod\ n)$. Therefore, the signature verification algorithm satisfies the correctness.

**The correctness of the confirmation algorithm:** The signer indicates that the message $M$ has been signed by the confirmation algorithm, and the final verifier needs to verify whether the following equation is established $R = H(M)^w \left( \dfrac{S_i}{\prod_{j=1,j\neq i}^{n} H(M)^{-e_j^2}} \right)^u (\bmod\ n)$.

Because the interaction phase has been calculated: $d_i = \log_{H(M)} \dfrac{S_i}{\prod_{j=1,j\neq i}^{n} H(M)^{-e_j^2}} (\bmod\ n)$ and $\dfrac{S_i}{\prod_{j=1,j\neq i}^{n} H(M)^{-e_j^2}} = H(M)^{d_i}(\bmod\ n)$, it is possible to verify the correctness of the algorithm, that is, to verify the equation. A non-signer cannot indicate that a message $M$ was signed by an confirmation algorithm, because the non-signer does not know $d_i$ and cannot generate $w$ that satisfies the equation $w = t - u \cdot d_i(\bmod\ n)$, *i.e.,* a non-signer can only make the equation established by guessing the value of $d_i$, $d_i \in Z_q^*$, then the probability that the non-signer indicates that the message was signed by the confirmation algorithm is $1/q$, which can be ignored.

## Theorem 2. If the probability of an adversary A winning in any polynomial time is negligible, then the deniable ring signature based on ISRSAC satisfies unforgeability

**Proof**: A simulator $S$ can be built to solve the big integer factorization problem with a non-negligible advantage, assuming that there exists a PPT opponent $A$ that is capable of forging signatures.

1. System establishment stage:
   (a) The simulator $S$ generates system parameters *params* and sends them to the adversary $A$;
   (b) The simulator $S$ initialize the lists: $List \leftarrow \emptyset$, $MList \leftarrow \emptyset$, $GSet \leftarrow \emptyset$;
   (c) The simulator $S$ sets $n$ ring members $\{A_1^*, A_2^*, \ldots, A_n^*\}$, so that each member's public key is $pk_i^* = (e_i, n_i)$, $i = 1, 2, \ldots, n$, and sends it to the adversary $A$.
2. Inquiry stage:
   (a) The adversary $A$ adaptively queries the oracle machine $Add(\cdot)$ and returns the user's public key $pk_i$;
   (b) The adversary $A$ adaptively queries the corrupt oracle machine $Crpt(\cdot)$ and returns the user's private key $sk_i$;
   (c) The adversary $A$ adaptively queries the signature oracle $DRSign(\cdot)$ to return the signature value $\sigma = Sign(m, sk_i, pk_1, \ldots, pk_n)$;
   (d) The adversary $A$ adaptive inquiry confirmation/denial oracle machine $C/D(\cdot, \cdot, \cdot)$.

Zhang et al. (2024), *PeerJ Comput. Sci.*, DOI 10.7717/peerj-cs.2190

3. Challenge stage: The adversary $A$ returns a forged signature $\sigma^*$, and for the same signed message and signer identity, if the adversary $A$ successfully forges an unqueried signature $\sigma^*$, the ring signature value $S_i$ in the ring signature $(S_1, \ldots, S_n)$ is generated by $S_i = H(M)^{d_i} \cdot \prod_{j=1, j \neq i}^{n} H(M)^{-e_j^2} \pmod{n}$ in the deniable ring signature scheme based on ISRSAC, and obviously the private key $d_i$ is required for the signing process. In the event that the adversary $A$ is successful in forging the ring signature, it will be imperative to ascertain the private key $d_i$ of the ring member $A_i$. Since the adversary $A$ relies on the challenging task of breaking large integers to crack private keys, the probability $\sigma^*$ of the adversary's successful forgery is small, meaning that the scheme satisfies the unforgeability requirement.

## Theorem 3. The deniable ring signature based on ISRSAC satisfies anonymity

**Proof**: The simulator $S$ generates system parameters *params* and sends them to the adversary $A$. The adversary $A$ adaptively queries each oracle machine, and sends the signed message $M^*$ and the public key ring $L^* = \{pk_1, \ldots, pk_n\}$ to the simulator $S$. The simulator randomly selects one $\pi (1 \leq \pi \leq n)$, calculates $\sigma_\pi \leftarrow RSign(M^*, n, L^*, sk_\pi)$, and sends $\sigma_\pi$ to the adversary $A$. $A$ guesses the corresponding value $\pi'$. If $\pi' \equiv \pi$, then the simulation succeeds; otherwise, the simulation fails.

Assuming that the adversary $A$'s computing power is unlimited and all publicly available parameter information in the scheme is known, adaptive access to all oracle machines is available. The simulator $S$ randomly selects a ring member $w$ to sign the message $m$. It can be seen $S_j = H(M)^{e_i e_j} \pmod{n}$ from the deniable ring signature based on ISRSAC algorithm that when w constructing each ring signature value, it is necessary to first randomly generate a public–private key pair that is distinguishable from the existing member, that is, the probability of the ring member generating all $S_i$ is $1/n, 1/n, \ldots, 1/n$ ($n$ in total), that is, the probability of any ring member generating the value of the ring signature is $1/n, 1/n, \ldots, 1/n$ ($n$ in total). Therefore, the probability that the adversary A can successfully distinguish the real signer among the members in the ring is not greater than $1/n$, and the probability of simulation success is negligible, that is, $Adv_{\Sigma,A}^{anon} = |2Pr[Exp_{\Sigma,A}^{anon-b} = b] - 1|$ is negligible, so it is proved that the deniable ring signature scheme based on ISRSAC satisfies anonymity.

## ANALYSIS OF EFFICIENCY

The deniable ring signature scheme proposed by *Komano et al. (2006)*, the lattice-based deniable ring signature scheme proposed by *Gao et al. (2018)*, the deniable ring signature scheme based on the elliptic curve discrete logarithmic assumption (EC-DL) proposed by *Zeng, Jiang & Qin (2012)*, and the proposed scheme of semi-trusted third-party anonymous identity authentication mechanism of VANETs (Vehicular Ad Hoc Network), a unique kind of wireless self-organizing network designed to provide communication between vehicles and between vehicles and infrastructure based on blockchain by *Jiang et al. (2021)*, are compared and analyzed below, which further illustrates the benefits of the proposed scheme.

**Table 2** Comparison of communication costs.

| Scheme | Algorithm | Sign | Confirm | Disavowal | Interactivity |
|---|---|---|---|---|---|
| *Komano et al. (2006)* | Zero-knowledge proofs | $(2n+1)|q|+|G|$ | $3|q|+3|G|$ | $(4|q|+|z|+2|G|)v$ | Yes |
| *Gao et al. (2018)* | Lattice | $2\ln m\log p+2\ln+(m+1)n\log p$ | $nm\log p$ | $2mn$ | No |
| *Zeng, Jiang & Qin (2012)* | EC-DL | $1p+4e+nm$ | $3p+3e+nm$ | $3p+6e$ | No |
| *Jiang et al. (2021)* | Bilinear pairs | $3p+4e+3nm$ | $2p+3e+nm$ | $6p+e+3m$ | No |
| Ours | ISRSAC | $n|q|$ | $4|q|$ | $3|q|$ | Yes |

## Analysis of communication overhead

We employ the following symbols in our experiments: The symbols $p$, $c$, $e$, $m$, $n$, and $k$ represent paring, elliptic curves, exponentiation, and scalar multiplications, respectively. The variables $|q|$, $|z|$, and $|G|$ represent the number of bits of the prime numbers $q$, $z$ and the number of bits of the components in group $G$.

*Komano et al. (2006)* first put up the idea of deniable ring signature, which is similar to a group signature scheme without the need for a group manager. However, the computational expense of signing and verifying is more than that of group signatures, and the signature algorithm's communication overhead is $(2n+1)|q|+|G|$.

Compared with the scheme of *Komano et al. (2006)*, the deniable ring signature scheme designed based on the SM2 national secret algorithm has less communication overhead in the signature algorithm; that is $(n+2)|q|+|G|$, which is more practical.

The deniable ring signature scheme designed by *Gao et al. (2018)* based on the lattice leads to a larger signature size due to the addition of repudiation, and the scheme can resist quantum attacks, so it has a certain price in terms of signature size, and has non-interaction and repudiation.

The conditional anonymous ring signature scheme proposed by *Zeng, Jiang & Qin (2012)* under the security model of *Komano et al. (2006)* is four times faster in the signing and verification stages, and has a constant cost because the scheme confirms and denies the protocol as non-interactive.

In order to significantly reduce the amount of time needed for communication, *Jiang et al. (2021)* introduced a lightweight anonymous authentication mechanism for VANETs based on the combination of ring signature and blockchain technology. User authentication simply needs to confirm whether the identity information is stored on the blockchain. The blockchain consensus technology simultaneously places the onus of privacy protection on users.

In the deniable ring signature based on ISRSAC algorithm proposed in this study, the output signature value is $\sigma = (S_1, S_2, \ldots, S_n)$, where $S_1, S_2, \ldots, S_n \in Z_q^*$. The communication cost of the signature algorithm is $n|q|$, that is, the communication cost of the signature algorithm has a linear relationship with the ring member. In the confirmation algorithm, the communication content is $t, R, u, w$, where $t, R, u, w \in Z_q^*$, that is, the confirmation algorithm communication overhead is $4|q|$. In a disavowal algorithm, the communication content is $x, u, y$, where $x, u, y \in Z_q^*$, that is, the disavowal algorithm communication overhead is $3|q|$. A comparative analysis of the above programs is shown in Table 2.

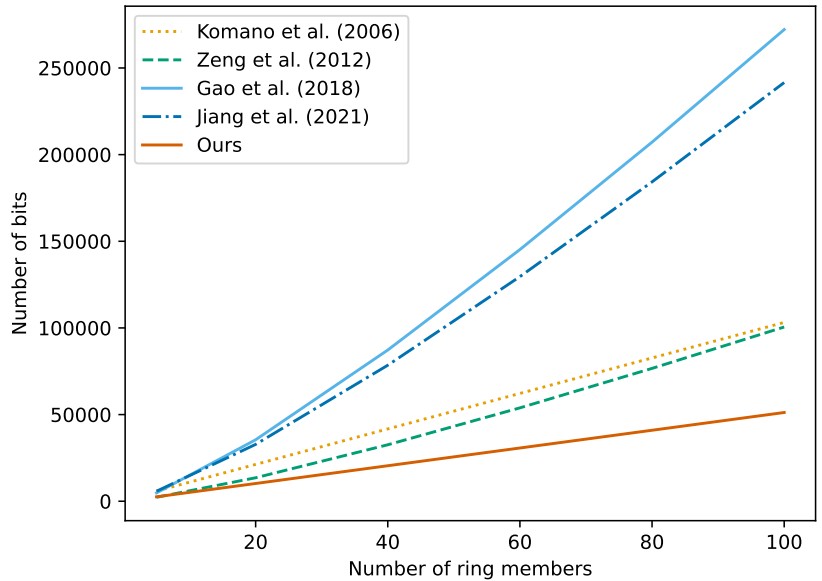

**Figure 1** Comparison of communication costs between different algorithms.

For visual comparison, we set $|q| = 512$ bits, $|G| = 256$ bits. The comparison of the signing communication overhead of each scheme is shown in Fig. 1.

## Analysis of computational overhead

In the analysis of computational overhead, three schemes of interactive confirmation/denial algorithms are selected for comparison. We use $T_{mul}$ to represent the point multiplication of elements on $G$; $T_{add}$ to represent the addition of dots of elements on $G$; $T_{H_1}$ means to execute the hash function $H_1 : \{0,1\}^n \rightarrow Z_q^*$; $T_{H_2}$ means to execute the hash function $H_2 : \{0,1\}^n \rightarrow G$; $T_{inv}$ represents the inverse operation performed on $Z_q^*$. The deny phase is set to round $v = 1$. The comparison of the computational overhead of the deniable ring signature scheme based on ISRSAC, the deniable ring signature based on SM2 and scheme *Komano et al. (2006)* is shown in Table 3.

In summary, the deniable ring signature based on ISRSAC has good practicality and is suitable for scenarios such as the anonymous authentication system. It can be used to implement an anonymous authentication system in which users can provide a ring signature to prove that they have an identity or authority while still being able to deny their own signature and protect personal privacy.

In digital asset transactions, anonymity and non-repudiation of transaction behavior are ensured. Trading participants can use a deniable ring signature to prove that they have sufficient funds to trade while also denying their own signature, preventing others from tracking the flow of their funds.

In secure communication, it can be used to ensure the confidentiality and non-repudiation of communications. The sender can sign messages using a deniable

**Table 3  Comparison of calculation costs.**

| Scheme | Algorithm | $T_{mul}$ | $T_{add}$ | $T_{H_1}$ | $T_{H_2}$ | $T_{inv}$ |
|---|---|---|---|---|---|---|
| ISRSAC-based | Sign | $n+1$ | – | $n$ | – | – |
| | Verify | $n$ | – | 1 | – | – |
| | Confirm | $n+2$ | 1 | 2 | – | – |
| | Deny ($v=1$) | $2+e/2$ | – | 1 | – | – |
| SM2-based | Sign | $4n-1$ | $2n-2$ | $n$ | 1 | 1 |
| | Verify | $4n$ | $2n$ | $n$ | 1 | 0 |
| | Confirm | 10 | 4 | – | – | – |
| | Deny ($v=1$) | $10+z/2$ | 6 | 2 | – | – |
| *Komano et al. (2006)* | Sign | $4n-2$ | $2n-2$ | 1 | – | – |
| | Verify | $4n$ | $2n$ | 1 | – | – |
| | Confirm | 10 | 4 | – | – | – |
| | Deny ($v=1$) | $10+z/2$ | 6 | 2 | – | – |

ring signature while denying its own signature, protecting the confidentiality of the communication and avoiding denial signatures by third parties.

In digital proof of stake, it can be applied to digital proof of stake, such as digital copyright protection. Copyright owners can use a deniable ring signature to prove that they own copyright to digital content while also being able to deny their own signature to prevent others from denying copyright attribution.

In summary, the deniable ring signature based on ISRSAC demonstrates good practicality and is suitable for scenarios such as anonymous authentication systems. It can be employed to establish an anonymous authentication system wherein users can furnish a ring signature to authenticate their identity or authority yet retain the ability to repudiate their own signature, thus safeguarding personal privacy.

## CONCLUSIONS

This article proposes a deniable ring signature scheme based on ISRSAC, which combines the anonymity and repudiation characteristics of ring signatures, which can provide privacy protection in scenarios such as electronic voting and regulatory blockchain and meet the needs of signer tracking at the same time. This technology allows individuals to vote or conduct transactions without revealing their identity. At the same time, it includes methods for identifying the actual participants if needed. This feature addresses the need for privacy protection and tracking of actual participants, especially in electronic voting and regulated blockchains. This article describes the specific algorithm process of the scheme implementation, proves the security of the deniable ring signature, compares and analyzes the communication cost and computational overhead of the proposed scheme and the current deniable ring signature scheme, and the results show that the proposed scheme has obvious advantages in communication overhead and computational overhead, which is applicable to practical scenarios.

In terms of specific implementation, the scheme is based on the ISRSAC signature algorithm, which has higher security than an improved RSA algorithm. The scheme makes

full use of the features of ISRSAC and combines the mechanism of deniable ring signatures to achieve the goal of satisfying both privacy protection and signer tracking.

From the practical point of view, the communication overhead and computing overhead are analyzed, and the results show that the proposed scheme has relatively low communication cost, which is beneficial for resource-constrained environments, and the low computational overhead of the scheme can have good practicability in practical applications.

Future research directions could include further optimizing the performance of deniable ring signature schemes, particularly in handling large-scale data and improving signature efficiency. Additionally, exploration could focus on extending this technology to scenarios involving multiple parties and further investigating its potential applications in privacy protection and identity verification to meet the growing demands in electronic voting and regulated blockchain domains. These studies will help further emphasize the importance and practicality of deniable ring signatures in information security and privacy protection.

### Funding
This research is supported by the Fundamental Research Funds for the Central Universities (No. 328202226) and the National Natural Science Foundation of China (No. 62002003). The funders had no role in study design, data collection and analysis, decision to publish, or preparation of the manuscript.

### Grant Disclosures
The following grant information was disclosed by the authors:
The Fundamental Research Funds for the Central Universities: No. 328202226.
The National Natural Science Foundation of China: No. 62002003.

### Competing Interests
The authors declare there are no competing interests.

### Author Contributions
- Yanshuo Zhang conceived and designed the experiments, performed the experiments, analyzed the data, performed the computation work, prepared figures and/or tables, authored or reviewed drafts of the article, and approved the final draft.
- Yuqi Yuan conceived and designed the experiments, performed the experiments, analyzed the data, performed the computation work, prepared figures and/or tables, authored or reviewed drafts of the article, and approved the final draft.
- Ning Liu conceived and designed the experiments, performed the experiments, analyzed the data, prepared figures and/or tables, authored or reviewed drafts of the article, and approved the final draft.
- Ying Chen conceived and designed the experiments, analyzed the data, authored or reviewed drafts of the article, and approved the final draft.

- Youheng Dong conceived and designed the experiments, analyzed the data, authored or reviewed drafts of the article, and approved the final draft.

## Data Availability

The raw data and the Python code used in this study are available in the Supplemental File.

## Supplemental Information

Supplemental information for this article can be found online at http://dx.doi.org/10.7717/peerj-cs.2190#supplemental-information.

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
