# Peer review of "Deniable ring signature scheme based on the ISRSAC digital signature algorithm"

_PeerJ Computer Science, doi:10.7717/peerj-cs.2190_

## Round 0.1 · original submission · Major Revisions

Please consider the comments from the two reviewers and revise the manuscript accordingly.

**Language Note:** PeerJ staff have identified that the English language needs to be improved. When you prepare your next revision, please either (i) have a colleague who is proficient in English and familiar with the subject matter review your manuscript, or (ii) contact a professional editing service to review your manuscript. PeerJ can provide language editing services - you can contact us at [email protected] for pricing (be sure to provide your manuscript number and title). – PeerJ Staff

·

Basic reporting

The manuscript uses a formal English style to convey professionalism and clarity. However, a close examination of the text reveals inconsistencies in the treatment of symbols, with some being italicized inconsistently. To ensure coherence and precision, a thorough review and standardization of symbol usage is required. Furthermore, the Introduction section would benefit from more references, particularly for key algorithms such as the ISRSAC algorithm, which are not properly cited. It is critical not only to supplement the references but also to ensure their relevance and currency, thereby maintaining the freshness of the scholarly discourse within the manuscript.

Experimental design

The paper's structure contains several logical inconsistencies, particularly in the Background section, where the presentation sequence should prioritize introducing challenging problems before delving into algorithms and security models. This reordering would give readers a better understanding of the contextual landscape before getting into technical details.
In addition, in order to increase understanding and confidence in the solution's efficacy, the security proof analysis accompanying each theorem must be enhanced. By elucidating the design principles that underpin the solution, readers can gain a better understanding of its robustness and reliability.
Furthermore, in the performance comparison analysis section, synthesis and generalization of the comparative results are required to effectively demonstrate the benefits of the proposed solution. Such an approach ensures clarity while emphasizing the solution's superiority over existing alternatives.

Validity of the findings

In the performance comparison analysis section, summarizing and generalizing the comparative results is critical for effectively highlighting the benefits of the proposed solution.
In the conclusion section, it should highlight the practical implications and performance benefits of the proposed approach. This will help enriching the paper's overall contribution to the field.

Additional comments

Check for formatting errors in the text, such as the incorrect formula in the second clause of Definition 2, which must be corrected.

Reviewer 2 ·

Basic reporting

- The manuscript appears to be written in a formal English style and is well organized.
- This paper focuses on deniable ring signatures. To demonstrate recent research progress on deniable ring signatures, the authors should review more relevant works in the manuscript.
- The research findings are effectively presented through the carefully designed tables and figures.
- However, the manuscript still requires significant improvement before it can be considered ready for publication. There are a few points listed below that require further explanation.

Experimental design

- What are the advantages of the author's proposed deniable ring signature scheme based on ISRSAC signature over existing schemes? When compared to current denial ring signature schemes based on SM2, the proposed scheme shows no significant improvement in terms of performance or other factors.
- What are the advantages and characteristics of the proposed solution in this article, and how do its functions apply to different usage scenarios? Please provide specific information.

Validity of the findings

- The authors claim their scheme is more secure than RSA-based schemes. Which section of the paper does this conclusion appear?
- The proofs of the theorem could be more detailed for better reader understanding.

Additional comments

- The symbols must be consistent throughout the entire text. In some places, DRSign(.,.,.) is used, but in others, it is written as DRSign(.;.,.). Careful verification and standardization are required.

Cite this review as

---

## Round 0.2 · Minor Revisions

Please revise the manuscript to address the final comments from the reviewers.

·

Basic reporting

- Although additional related studies have been included and discussed in this revision, I believe that further literature support from related fields could still be incorporated.

Experimental design

- The synthesis and generalization of the comparative results have been enhanced in this revision.

Validity of the findings

- While the practical implications and performance benefits of the proposed approach are included in the conclusion, it seems somewhat redundant and would benefit from being trimmed down.

Reviewer 2 ·

Basic reporting

- The authors may want to revise the abstract because it contains information that is not entirely irrelevant to the research.
- The conclusion can be strengthened by emphasizing future research directions and the significance of the study.

Experimental design

- This section has been improved with this revision. I don't have any more comments.

Validity of the findings

- The legends in Figure 1 should be revised so that the names of the methods in this figure are more consistent with those in Tables 2 and 3.

Additional comments

- To ensure that academic citation standards are followed, the authors should check the references for completeness and consistency.

Cite this review as

---

## Round 0.3 · accepted · Accept

The authors have addressed all of the reviewers' comments, which were quite minor. I have evaluated the revision, and I am satisfied with the current version. In my opinion, the manuscript is ready for publication.